# CD6 and Its Interacting Partners: Newcomers to the Block of Cancer Immunotherapies

**DOI:** 10.3390/ijms242417510

**Published:** 2023-12-15

**Authors:** Lucía Aragón-Serrano, Laura Carrillo-Serradell, Violeta Planells-Romeo, Marcos Isamat, María Velasco-de Andrés, Francisco Lozano

**Affiliations:** 1Institut d’Investigacions Biomèdiques August Pi i Sunyer (IDIBAPS), Rosselló 149-153, 08036 Barcelona, Spain; laragose79@alumnes.ub.edu (L.A.-S.); lcarrillo@recerca.clinic.cat (L.C.-S.); planellsi@recerca.clinic.cat (V.P.-R.); mvelascod@recerca.clinic.cat (M.V.-d.A.); 2Sepsia Therapeutics S.L., 08908 L’Hospitalet de Llobregat, Spain; misamat@sepsia.com; 3Servei d’Immunologia, Centre de Diagnòstic Biomèdic, Hospital Clínic de Barcelona, 08036 Barcelona, Spain; 4Departament de Biomedicina, Facultat de Medicina, Universitat de Barcelona, 08036 Barcelona, Spain

**Keywords:** cancer, immunotherapy, CD6, scavenger receptors, CD166/ALCAM, CD318/CDCP1, CD44, decoy receptor, adoptive cell transfer, chimeric antigen receptor, monoclonal antibodies

## Abstract

Cancer management still requires more potent and safer treatments, of which immunomodulatory receptors on the lymphocyte surface have started to show promise in new cancer immunotherapies (e.g., CTLA-4 and PD-1). CD6 is a signal-transducing transmembrane receptor, mainly expressed by all T cells and some B and NK cell subsets, whose endogenous ligands (CD166/ALCAM, CD318/CDCP-1, Galectins 1 and 3) are overexpressed by malignant cells of different lineages. This places CD6 as a potential target for novel therapies against haematological and non-haematological malignancies. Recent experimental evidence for the role of CD6 in cancer immunotherapies is summarised in this review, dealing with diverse and innovative strategies from the classical use of monoclonal antibodies to soluble recombinant decoys or the adoptive transfer of immune cells engineered with chimeric antigen receptors.

## 1. Introduction

Cancer is an unmet clinical need responsible for high morbidity and mortality rates worldwide, which is among the main causes of premature death in more than 100 countries. Estimates of the Global Cancer Observatory (gco.iarc.fr) determined that the most diagnosed cancer in 2020 was female breast cancer (11.7% of total cases) for both sexes, but lung cancer was the leading cause of cancer death (18.0% of total cancer deaths) [1]. It is estimated that in 2040, 28.4 million new cancer cases will occur worldwide, increasing the number of cases in 2020 by 47% (19.3 million) [1]. These projections make the development of highly effective therapies with minimal toxicity peremptory.

Large differences between the existing types of cancer have precluded a universal therapy. Cancers can be classified into two general groups: haematological cancers, which consist of hyperproliferative blood cells or precursors, and solid tumours, in which the hyperproliferative cells are found in solid organs. The most widely used therapies against both types of cancer have been conventional surgery, chemotherapy, and radiotherapy. For some solid tumours such as hepatocellular carcinoma, surgery is the best treatment option [2]. However, the outcome may lead to recurrence in cases of late detection. On the other hand, chemotherapy and radiotherapy eliminate duplicating cells by different mechanisms, but they present low specificity and important side effects such as nephrotoxicity, neurotoxicity, or cardiovascular complications, among others [3,4]. Moreover, radiotherapy can cause healthy cells to initiate carcinogenesis processes and chemotherapy can generate drug resistance. This scenario prompts the development of alternative and adjunctive treatments with the aim of increasing specificity and decreasing off-tumour cytotoxicity [5].

Recent breakthroughs in cancer treatment have come from the field of immunotherapy. Classical cancer immunotherapy relies on cytotoxic monoclonal antibodies (mAbs) against tumour-specific antigens (TSAs). This has evolved into the generation of soluble decoy receptors, mainly in the form of fusion proteins linked to the Fc portion of human IgG [6]. However, the most currently extended immunotherapy is the blockade of immune checkpoint inhibitors (ICI). This strategy specifically aims at interfering with the interaction between PD-1 and CTLA-4 with their ligands (PD-L1/2 and B7.1/2, respectively), thus avoiding T-cell exhaustion and potentiating ongoing anti-tumour immune responses [7]. Though some studies report greater safety than chemotherapy [8], the risk of developing immune-related adverse events of different intensities is hindering its complete success [9]. Targeted immunotherapies, such as adoptive transfer of immune cells engineered to express chimeric antigen-specific receptors (CARs), have also gained recent interest due to its efficacy against haematological malignancies [10]. Its performance against solid tumours is obstructed, however, by their intrinsic heterogeneity and the presence of hostile immunosuppressive microenvironments [11]. The present review will summarise recent reports on humoral and cellular anti-tumour strategies targeting CD6, a long known but less studied immunomodulatory lymphocyte receptor [12]. The multifaceted properties of CD6 make this receptor an exciting candidate for its use as an alternative and adjunctive immunotherapy in cancer.

## 2. The CD6 Receptor: Structure, Function, Tissue Distribution, and Ligands

CD6 is a type I transmembrane glycoprotein of 105–130 kDa from the ancient and highly conserved Scavenger Receptor Cysteine-Rich (SRCR) superfamily [13]. CD6 is expressed by all T cells, a subset of B (B1a) and NK (CD56^dim^CD16^+^) cells, as well as some haematopoietic precursors and certain regions of the central nervous system [14,15,16,17]. CD6 is composed of an extracellular region, which consists of three tandem SRCR domains, a transmembrane region, and a cytoplasmic tail devoid of intrinsic catalytic activity but harbouring several phosphorylatable residues (Ser, Thr, and Tyr) suitable for intracellular signal transduction [17,18] (Figure 1A). Accordingly, CD6’s cytoplasmic tail encompasses pro-rich motifs that can interact with SH3 domain-containing proteins, Tyr-based motifs that can interact with SH2 domain-containing proteins, and Ser/Thr-rich motifs targeted by Ser/Thr kinases [18,19]. The long list of intracellular signal transducers reported to interact either directly or indirectly with CD6 include both enzymes (LCK, FYN, ZAP70, ITK, SHP1, RasGAP, CK2, and PKC) and protein adaptors (SLP76, TSAD, GADS, GRB2, and syntenin) [20]. Aside from the prototypical membrane-bound CD6, a circulating soluble CD6 form (sCD6), resulting from proteolytic cleavage by metalloproteases following T-cell activation, has also been reported [21,22] and whose physiological role is uncertain.

CD6 physically associates with the TCR/CD3 complex in resting T cells and co-localises with it at the centre of the immunological synapse, where it helps to stabilise the adhesive contacts with antigen-presenting cells (APC) and modulate subsequent TCR/CD3 intracellular signals driving to proliferative and differentiation responses [23,24,25] (Figure 1B). Early in vitro studies with different anti-human CD6 mAbs supported a co-stimulatory function for CD6 in human T-cell activation by potentiating the effect of polyclonal T-cell activators such as anti-CD3 or phorbol esters (PMA, phorbol-myrisitate acetate) [26,27,28,29]. This notion was, however, contradicted by later in vitro and in vivo studies positioning CD6 as a negative modulator of T-cell activation [28,29,30]. First evidence on this regard was the observation that human T cells expressing rat CD6 show lower levels of calcium mobilisation in response to superantigen exposure compared with T cells not expressing the exogenous receptor [28,29,30] Definitive evidence was obtained as soon a CD6-deficient mice (*Cd6^−/−^*) were made available [28,29,30]. Thus, *Cd6^−/−^* mice of different genetic backgrounds (C57BL/6N and DBA-1) showed higher proliferative responses than wild-type ones to in vitro anti-CD3 mAb stimulation, a fact compatible with attenuation of TCR/CD3 signalling by CD6 [28,29,30]. Intriguingly, the same mice showed either attenuated or exacerbated phenotypes when subjected to experimental models of autoimmune disease (experimental autoimmune encephalitis, EAE; collagen-induced arthritis, CIA; experimental autoimmune uveitis, EAU; imiquimod-induced psoriasis; and chronic graft versus host-induced lupus-like), a fact that was dependent on the mouse background and the experimental model [28,29,30,31,32,33]. Recent quantitative mass spectrometry studies have conciliated the two views by showing that the CD6 signalosome constitutes a multitask hub that contributes to the diversification of TCR/CD3 signalling by recruiting both activating (SLP-76, ZAP70, and VAV1) and inhibitory (UBASH3A/STS-2) intracellular signalling mediators [34]. On this basis, it can be speculated that the stimulation conditions would determine the relative abundance of such recruited mediators, giving rise to different (attenuated/exacerbated) in vitro and in vivo T-cell responses.

CD6’s involvement in lymphocyte activation, proliferation, and survival processes is achieved via the interaction with its endogenous ligands. The best characterised one is CD166/ALCAM (for Activated Leukocyte Cell Adhesion Molecule), a 100 kDa adhesion molecule of the immunoglobulin superfamily (IgSF) with a broad tissue distribution including activated mature T and B lymphocytes, melanoma cells, fibroblasts, bone marrow cells, endothelial cells, cerebral cortex, and epithelial cells from skin, thymus, digestive tract, pancreas, liver, and kidney [35,36]. Interestingly, CD166/ALCAM is overexpressed in different cancer types and has been involved in some tumour-related processes, such as tumour progression or poor prognosis [37]. The CD6-CD166/ALCAM interaction is established between the membrane-proximal SRCR domain (D3) of the former and the amino-terminal domain (V1) of the latter [38]. This heterophilic interaction has 10 to 100 times more affinity than the cis- and trans-acting homophilic CD166/ALCAM-CD166/ALCAM adhesive interactions [39] and plays a relevant role in several aspects of T-cell physiology such as thymocyte development [29,40], T-APC immunological synapse stabilisation [23,24,25], and T-cell transmigration to inflamed tissues [41,42]. Regarding the latter, a very recent report shows that upon CD6 binding, CD166/ALCAM and other CD6’s ligands (CD318/CDCP1 and CD44; see next paragraph) cluster at the membrane of human epithelial cells, triggering actomyosin cytoskeleton remodelling and subsequent disassembly of the tight junctions responsible for epithelial barrier integrity [43]. On this basis, the authors claim this as the mechanism by which CD6 would mediate T-cell-driven disruption of tissue barriers during inflammation.

Other *bona fide* CD6 endogenous ligands include Galectins 1 and 3 [44], CD318/CDCP1 (for CUB domain-containing protein 1) [45], and CD44 [43]. Galectins 1 and 3 are soluble glycan-binding proteins of broad tissue distribution (e.g., immune cells, sensory neurons, and endothelial and epithelial cells) [46] that regulate essential physiological processes, such as cell adhesion, migration, cytokine synthesis, and survival. Abnormal expression of Galectins 1 and 3 is often found in cancers and is associated with the development, progression, and metastasis of different types of cancer [47]. Galectins 1 and 3 bind to both CD6 and CD166/ALCAM and interfere with superantigen-induced T-cell proliferation and cell adhesion phenomena mediated by heterophilic CD6-CD166/ALCAM interactions but not homophilic CD166/ALCAM ones. Moreover, CD6 expression protects T cells from Galectin 1- and 3-induced apoptosis [44]. The mechanism by which CD6 surface expression negatively modulates Galectin 1- and 3-induced T-cell death has not been explored in depth yet. Studies in a CD6-negative Jurkat T-cell derivative (2G5 cells) transfected with wild-type and cytoplasmic tail-truncated CD6 isoforms indicate that such down-modulatory effect depends on the integrity of the CD6’s cytoplasmic domain [44]. This could relate to the previously reported fact that CD6 ligation protects B-type chronic lymphocytic leukemia (B-CLL) cells from IgM-induced apoptosis by inducing an increased bcl-2/bax_α_ ratio [48]. As for CD318/CDCP1 (also known as TRASK, SIMA135, or gp140), this CD6 ligand is expressed by fibroblasts and epithelial cells with which T cells interact but not by immune cells [45]. CD318 (CDCP1) is found overexpressed by epithelial and myeloid cancer cells, where it correlates with cancer progression and metastasis initiation [49]. Very recently, biochemical and biophysical evidence support the interaction of CD6 with CD44 [43], a broad tissue distribution and multifunctional cell adhesion receptor mainly involved in hyaluronic acid recognition, the major extracellular matrix component. Different CD44’s isoforms are expressed in various cancer types, which serve as prognostic biomarkers and therapeutic targets [50].

In light of the above, CD6 can be considered a multifaceted lymphocyte receptor. Thanks to its intrinsic intracellular signalling capabilities, CD6 can modulate either positively or negatively the lymphocyte (T, B1a, NK) functions in different immune response contexts. How the signalling capabilities of CD6 contribute to different immune response outputs is not fully understood. As stated above, recent proteomic data shows that the CD6 signalosome behaves as a hub that assembles many different enzymes and adaptors, which can impact positively or negatively on signal propagation in different stimulation contexts and potentially drive to divergent immune response outcomes [34]. On the other side, given its heterophilic interactions with cell-bound and soluble endogenous ligands, CD6 can modulate cell-to-cell contacts involved in the initiation and maintenance of immune response in different contexts. This includes anti-tumour and the autoimmune immune responses, which are two sides of the same coin [49].

## 3. Depleting and Non-Depleting Anti-CD6 mAb-Based Strategies

Many mAbs are currently in use as therapeutic drugs for a large number of immune-related clinical disorders [6]. This is the case of cancer, where mAbs are used to favour direct or indirect purging of tumour cells by means of diverse mechanisms, triggering antibody-associated responses like antibody- dependent cell cytotoxicity (ADCC), antibody- dependent cell phagocytosis (ADCP), and a blockade of receptor–ligand interactions, thereby interfering with cell signalling pathways and modulating the tumour microenvironment, among others [51].

The first anti-human CD6 mAb available was 12.1, which recognised an antigen (T12) present on nearly all T cells and the majority of B-CLL cells [14]. Soon after, the mouse 12.1 mAb was used as an immunosuppressive agent in human kidney and bone marrow transplantation to prevent acute rejection based on its T-cell-depleting cytotoxic properties [52,53] (Table 1). Subsequently, the original mouse version (IOR-T1) of itolizumab, a humanised mAb against human CD6 currently approved for the treatment of psoriasis [54] (Table 1), was assayed for topical treatment of cutaneous T-cell lymphoma (CTCL) skin lesions [55] (Table 1).

The humanised version of the non-depleting mouse anti-human CD6 UMCD6 mAb previously tested in experimental mouse models of autoimmune encephalitis and arthritis [30,66] has recently been tested in pre-clinical models of human T-cell lymphoma (TCL) [57] (Table 1). To that end, the humanised UMCD6 mAb was conjugated to monomethyl auristatin E (MMAE), an FDA-approved anti-mitotic toxin, for CD6 internalisation and subsequent killing of CD6-positive cells (Figure 2A, left-hand side). In contrast to the unconjugated mAb, the antibody-drug conjugate (CD6-ADC) selectively killed TCL cells in vitro and, more importantly, its systemic or local administration significantly reduced established TCL tumours in immunodeficient NSG mice. Interestingly, the authors also demonstrated that CD6-ADC did not kill normal primary CD6-expressing human T and NK human cells or other actively proliferating human cells not expressing CD6, thus reinforcing the specificity of the therapy [57]. These results support CD6-ADC’s therapeutic potential in fatal CD6-positive lymphoid neoplasms, albeit further pre-clinical studies are warranted to confirm its efficacy and toxicity profile.

Recent experiments in which human CD166/ALCAM- and/or CD318/CDCP1-positive cancer cell lines of different origins (i.e., breast, lung, and prostate) were co-cultured with human peripheral blood mononuclear cells (PBMCs) in the presence of the UMCD6 (anti-human CD6) or 3a11 (anti-human CD318) mAbs have rendered interesting, though somehow unexpected, results [56]. The authors observed that both mAbs enhanced in vitro killing of cancer cells by PBMCs. Nevertheless, PBMCs pre-incubated with the UMCD6 mAb were more effective than pre-incubated with the 3a11 mAb or the ICI blocker, pembrolizumab and nivolumab (two anti-human PD-1 mAbs), in increasing cancer cell death and lowering cancer cell survival in vitro [56]. The UMCD6 mAb also enhanced PBMCs’ killing of cancer cells in vivo as demonstrated in a xenograft mouse model of triple-negative breast cancer cells subcutaneously implanted in immunodeficient SCID mice [56] (Table 1). Mechanistically, the authors demonstrated that following UMCD6 mAb-induced CD6 internalisation in T and NK cells, a downregulated expression of the inhibitory NKG2A receptor and an upregulated expression of the activating NKG2D receptor, as well as of perforin and granzyme B, is observed in both NK and CD8^+^ T cells, thus resulting in increased anti-tumour cytotoxic capabilities (Figure 2B, left-hand side). Compared to ICI blockers, the enhanced anti-tumour response would be achieved without instigating serious autoimmune side effects since pre-clinical studies in mouse models of autoimmune disease (i.e., EAE, CIA, and EAU) show that UMCD6 and other anti-CD6 mAbs reduce clinical signs of disease, pathogenic CD4^+^ Th1/Th17 responses, and inflammatory cell infiltration into the target organs [30,32,56,66].

## 4. CD6 Decoy Receptor-Based Strategies

The use of soluble proteins as decoy receptors is a well-known strategy for blocking receptor–ligand interactions and a good source of new therapies for inflammatory disorders and cancer. Indeed, they constitute the second-most common class of agents among the approved ligand-targeting drugs in clinical trials, most of which are in the form of Fc-fusion proteins [6]. One of the first decoy receptors identified was the interleukin-1 type II receptor (IL-1RII), which was followed by others binding to members of the tumour necrosis factor (TNF) superfamily such as the TNF-related apoptosis-inducing ligand (TRAIL), the Fas ligand (FasL), and TNF superfamily members 14 (TNFSF14/LIGHT) and 15 (TNFSF15/TL1A) [67]. The mechanisms involved in the generation of mammalian decoy receptors include proteolytic cleavage of cell surface receptors (ectodomain shedding), phospholipase C-mediated cleavage, and alternative mRNA splicing or intronic polyadenylation [68].

As stated above, a circulating soluble form of the human CD6 (shCD6) form is found at low (pico/nanomolar) concentrations in healthy individuals, as opposed to higher concentrations found in individuals suffering from inflammatory disorders [21,22,69,70]. Such a naturally occurring shCD6 form is exclusively composed of by the whole extracellular region of CD6 (SCRC domains 1, 2, and 3). Due to its low serum availability, a recombinant protein identical to shCD6 was produced and purified from a mammalian expression system and used for further functional investigations [23,24,25] in vitro assays, showing that such recombinant human shCD6 inhibits T-cell proliferation comparably to anti-human CD6 and CD166/ALCAM mAbs [23,24,25], thus opening the possibility to new potential immunomodulatory uses (Table 1). Interestingly, pre-clinical studies exploring the immunomodulatory properties of shCD6 greatly benefit from the well-known interspecies conservation of the receptor–ligand interactions mediated by CD6, implying that the interaction of mouse and human CD6 with mouse or human CD166/ALCAM are interchangeable [71].

First evidence for the immunotherapeutic potential of shCD6 against cancer comes from transgenic mice expressing high circulating levels of human sCD6 (shCD6LckEμTg), which showed reduced tumour growth when challenged with subcutaneous syngeneic cancer cells of different lineages (i.e., melanoma B16.F10, fibrosarcoma MCA205, and lymphoma RMA-5) [60]. Similar lineage-independent enhanced anti-tumour responses were also observed in wild-type mice undergoing either repeated (every-other-day) infusions of the recombinant shCD6 protein or transduced with hepatotropic adeno-associated virus (AAV) coding for mouse sCD6 (smCD6) [60]. In vitro and in vivo functional mouse lymphocyte analyses found that the mechanisms underlying such a shCD6-mediated enhanced response were multiple, acting at both the lymphocyte effector function and tumorigenesis sides. From the former, they included the defective generation and the functionality of mouse regulatory T (T_reg_) cells, as well as impaired Galectin 1/3-induced mouse T-cell apoptosis. Regarding tumorigenesis, they included interference of homotypic CD166/ALCAM-CD166/ALCAM interactions needed for optimal proliferation and migration of mouse tumour cells [60] (Figure 2A,B, right-hand side). Whether shCD6 would interfere with the function of other CD166/ALCAM-expressing immunosuppressive cells from the tumour microenvironment (e.g., mesenchymal stromal cells) should be further addressed [72]. In line with previous studies using non-depleting anti-human and mouse CD6 mAbs [30,32,66], available evidence argues against the possibility that shCD6-based anti-tumour strategies would instigate the development of autoimmune side effects. This is supported by the observation of less severe outcomes from experimental models of arthritis (CIA) and encephalitis (EAE) in both transgenic (shCD6LckEμTg) and shCD6 protein-treated wild-type mice [31,60].

## 5. Adoptive Immune Cell Transfer-Based Strategies

Adoptive transfer of immune cells engineered to express CARs has emerged as a novel potent immunotherapy in cancer and beyond [10,73]. CARs typically consist of an extracellular target-binding domain, a hinge region, a transmembrane region, and a cytoplasmic region that contains one or several activation domains [10]. The extracellular region, usually harbouring the single-chain variable fragment (scFv) of a mAb, provides the antigen specificity. Despite the six CAR-T therapies currently approved by the American and European drug and medicine agencies (FDA and EMA, respectively) for haematological malignancies, no therapies have been approved against solid tumours, as they present more limitations [74].

The lack of tumour antigen specificity and tumour heterogeneity are important obstacles for CAR-based therapies in cancer, leading to severe on-target off-tumour toxicities and insufficient efficacy. The ideal targets for cancer therapies are tumour-specific antigens (TSA), which are found on cancer cells only but not on healthy cells. TSA are neoantigens that can result from oncogenic viral infections and tumour-specific somatic mutations (normally single nucleotide variants but also frameshifts, splice variants, gene fusions, or endogenous retroelements) [75]. TSA can be common across different cancer patients and not present in the normal genome (shared TSA), thus having the potential to be used as broad-spectrum anticancer therapies. In other cases, TSA are unique and completely different from patient to patient (personalised TSA) and thus can only be specifically targeted to each patient [76]. Alternative to TSA are tumour-associated antigens (TAA), which are found in high levels on cancer cells but are also expressed at lower levels on healthy cells. The CD19, CD22, and BCMA (B-cell maturation antigen) cell surface receptors are good examples of TAA for B-cell lineage malignancies; while the CEA (carcinoembryonic antigen), GPC3 (glypican-3), and PSA (prostate-specific antigen) ones are for solid tumours of colorectal, hepatic, or prostatic origin, respectively [74].

The most studied CD6’s ligand, CD166/ALCAM, is overexpressed in many haematological and solid tumours and thus could be considered as a TAA. So, designing CARs targeting this cell surface molecule could be an effective anti-cancer immunotherapeutic strategy, as demonstrated by a pioneer study with human T cells engineered to express an anti-human CD166/ALCAM CAR (CD166.BBζ-CAR) against human osteosarcoma cell lines [59] (Figure 2A, right-hand side; Table 1). CD166.BBζ-CAR T cells showed increased cytotoxicity to CD166/ALCAM-positive human osteosarcoma cells, but not to CD166/ALCAM-negative cells compared to non-transduced T cells, showing a correlation between CD166/ALCAM expression and cytotoxicity, and thus, specificity [59]. A large amount of tumour necrosis factor alpha (TNF-α) and interferon gamma (INF-γ) cytokines, as well as a more moderated amount of IL-2, were released in the culture medium in a CD166/ALCAM expression-dependent manner, though CAR T cells also produced high amounts of Th2 cytokines (IL-4, IL-6 and IL-10) that could bias the immune response [59]. The intravenous infusion of human CD166.Bζ-CAR T cells to immunodeficient mice (NOD/SCID) challenged with human osteosarcoma cells showed increased tumour infiltration, suppression of tumour growth, and lack of morphological changes in different organs (lung, heart, liver, spleen, intestine, and kidney) compared with non-transduced T cells and non-treated controls [59]. The observation of no obvious side effects in the immunodeficient mouse model used, which lack a functional immune system, still deserves further reassessment of on-target off-tumour toxicity issues.

As an alternative to anti-CD166/ALCAM-CAR T cells, the use of the extracellular region of human CD6 as an antigen-binding domain may also be a good strategy (Figure 2A, right-hand side; Table 1). As mentioned above, overexpression of endogenous CD6 ligands (i.e., CD166/ALCAM, CD318/CDCP1, and CD44) is associated with tumour events, so the use of CD6-based CARs may broaden the spectrum of treatable tumours. This view has been challenged by a recent in vitro study with human colorectal cancer cells (CRC), in which both CD166/ALCAM and CD318/CDCP1 are highly expressed [58]. To this end, a second generation CD6-based CAR was generated in which the antigen-binding domain was the whole human CD6’s extracellular region (CD6-CAR). Unexpectedly, human T cells expressing the CD6-CAR showed cytotoxicity effects only against CD166/ALCAM-positive human CRC-derived cell lines, inferring that the cytotoxic activity correlated with CD166/ALCAM but not CD318/CDCP1 expression [58]. The same results were observed when cytotoxicity assays were performed with HEK 293T transfectants overexpressing CD166/ALCAM or CD318/CDCP1 [58]. Moreover, the levels of INF-γ in supernatants of CD6-CAR T-cells’ co-cultures were significantly higher compared to the control co-cultures (CD19-CAR T cells) [58]. Of particular interest was the finding that CD6-CAR T cells also showed potent cytotoxic responses against human CRC cancer stem cells, which exhibit robust CD166/ALCAM expression.

Unpublished preliminary in vitro observations made by our group using human leukemic NK cell-derived KHYG-1 cells stably transduced with a second-generation CD6-CAR also show superior cytotoxic activity against high CD166/ALCAM-expressing human cancer cell lines from colon (DLD-1) and ovary (SKOV-3) origin, compared to the control non-transduced cells. In contrast, no cytotoxicity differences were observed between CD6-CAR-transduced and -non- transduced KHYG-1 cells when co-cultured with low/negative CD166/ALCAM expressing human lymphoblastoid (Daudi and Raji) and erythromieloid (K562) cell lines. Though in vivo efficacy and toxicity studies are still to come, the dependence on surface CD166/ALCAM dosage may help minimise on-target off-tumour toxicity. Moreover, the use of CAR NK cells is currently being explored for their advantages regarding CAR T cells. Among them, the use of off-the-shelf allogeneic NK cells and the lower probability of developing cytokine release syndrome (CRS) following CAR cell administration [77].

## 6. Concluding Remarks

All pre-clinical evidence available sustains promise for humoral and cellular anti-tumour strategies targeting CD6 against haematological and solid cancers, in its capacity to act as or mimic an immunomodulatory lymphocyte receptor. This would not only result from CD6 expression by T (TCL) and B (CLL) lymphoid malignancies, but also from the functional relevance of CD6–ligand interactions in anti-tumour immune responses and tumorigenesis. Much of this game is played via the selective expression of *bona fide* CD6 ligands (CD166/ALCAM, CD318/CDCP1, CD44, and Galectins 1/3) by cancer cells of many lineages, adding to tumour versus normal tissue specificity and offering the opportunity to readily target these ligands therapeutically while minimising on-target off-tumour toxicity. Current immunotherapies to potentiate the host’s anti-tumour immune responses via the blockade of immune checkpoint inhibitors and the adoptive transfer of engineered CAR cells constitute high hopes in cancer treatment but remain at risk of immune-related adverse events. These therapeutic approaches are nonetheless mostly unresponsive against solid tumours as their efficacy is largely obstructed by tumour heterogeneity and the tumour’s immunosuppressive microenvironment. In this respect, the multifaceted properties of CD6 and its ligands can be used to leverage its tumour specificity, infiltration capacity, immunomodulatory potential, and the on-target off-tumour toxicity observed in novel immunotherapeutic approaches, turning CD6 into an exciting candidate for alternative and adjunctive cancer immunotherapies.

## Figures and Tables

**Figure 1 ijms-24-17510-f001:**
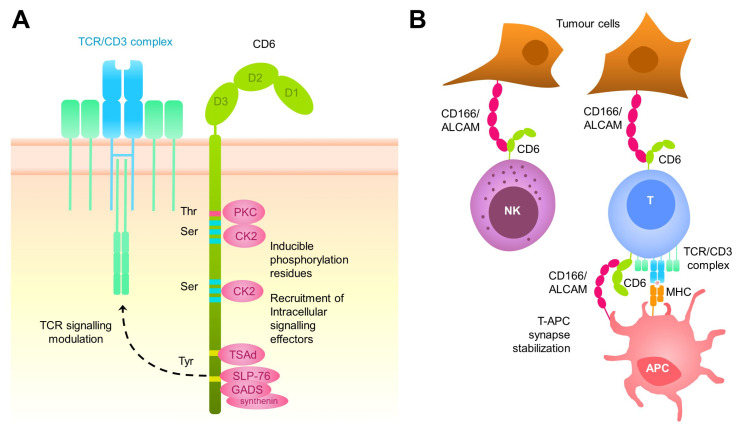
(**A**) Schematic representation of membrane-bound CD6 structure and intracellular interactions. (**B**) Cell-to-cell adhesive contacts involving CD6-CD166/ALCAM interactions during T and NK cell recognition of target cells.

**Figure 2 ijms-24-17510-f002:**
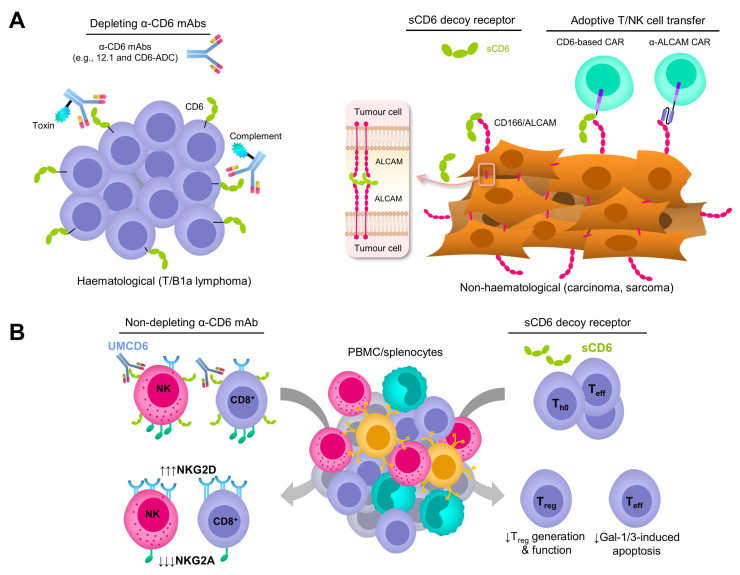
Schematic representation of different CD6-based strategies in cancer immunotherapy. (**A**) Strategies involving direct anti-tumour effects on CD6- and CD166/ALCAM-expressing tumour cells. (**B**) Strategies involving indirect anti-tumour effects resulting from modulation of immune cell generation and function.

**Table 1 ijms-24-17510-t001:** Summary of molecules (mAbs and engineered protein receptors) targeting CD6 and its interacting partners in pre-clinical and clinical immunotherapy studies.

Molecule	Type	Study Type	Disease	References
UMCD6	Humanised mouse anti-human CD6 mAb	Pre-clinical	CD166/ALCAM-expressing human tumours	[56]
CD6-ADC	Humanised mouse anti-human CD6 mAb (UMCD6) toxin-conjugated	Pre-clinical	Human T-cell lymphomas	[57]
CD6-CAR	CD6-based CAR-T cells	Pre-clinical	Human Colon adeno-carcinoma	[58]
CD166-CAR	Anti-human CD166 CAR-T cells	Pre-clinical	Human Osteosarcoma	[59]
IOR-T1	Mouse anti-human CD6 mAb	Clinical	Human cutaneous T-cell lymphoma (CTCL)	[55]
shCD6	Soluble human CD6 protein	Pre-clinical	Mouse tumours of different lineages	[60]
Anti-T12 (12.1)	Mouse anti-human CD6 mAb	Clinical	Human kidney transplant rejection	[53]
			Human bone marrow transplant rejection	[52]
Itolizumab	Humanised mouse anti-human CD6 (IOR-T1) mAb	Clinical (CTRI/2009/091/ 001009)	Psoriasis	[61,62,63]
		(RPCEC00000007 and RPCEC00000035, Cuban Registry of Clinical Trials)	Rheumatoid arthritis	[64,65]

## Data Availability

Data sharing not applicable—no new data generated.

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
