# Peer review of "CD6 and Its Interacting Partners: Newcomers to the Block of Cancer Immunotherapies"

_ijms, 2023, doi:10.3390/ijms242417510_

Round 1

Reviewer 1 Report

Comments and Suggestions for Authors

Within this short review, the authors summarize the potential and use of anti-CD6 antibodies for the immunotherapy of human malignencies. It is well written and most of the content is clearly presented. However, I still have some comments/requests.

A figure summarizing the role of CD6 in a "classical" / tumor immune response (and its signalling) would be helpful.

Please clarify the paragraph about Galectin1/3 interaction and CD6 (lines 90 to 100): What I get is: Gal1/3 are ligands of CD6, but how does CD6 inhibit the galectin1/3 induced cell death (because it has a higher affinity and lead to a not-binding to the death inducing receptor? If yes, please add a a sentence for explanation. thank you

In this paragraph, I miss also a description of the signalling properties of CD6. How does this contribute to the multifacetted role? 

The last part (5) focusses more on CD166; maybe one shoudl ammend the title saying (CD6 and its binding partner CD166) and add also some (maybe) more general information on CD166. Or just "CD6 and its binding partners".

For me it was not really clear which clinical studies are performed with anti-CD6 molecules. Maybe you could also add the registration numbers for the studies and/or a table listing them.

In general: I am missing a lot of the murine studies, in which the function and potency of CD6 has been studied in detail.

Please state more clearly in the respective paragraphs, if you are summarizing effect/functions/expression data of the human or the mouse system.

Please improve the resolution of the provided figure, it looks "blurry".

Comments on the Quality of English Language

There are some minor misspellings and missing commata. Please correct :-)

Author Response

Referee #1

Comments and Suggestions for Authors

Within this short review, the authors summarize the potential and use of anti-CD6 antibodies for the immunotherapy of human malignencies. It is well written and most of the content is clearly presented. However, I still have some comments/requests.

A figure summarizing the role of CD6 in a "classical" / tumor immune response (and its signalling) would be helpful.

Reply: Following the reviewer’s indication a new figure has been included in the revised version of the manuscript (new Figure 1) summarizing (A) important structural and functional properties of the membrane-bound form of CD6 and (B) cell-to-cell adhesive contacts mediated by the CD6-CD166/ALCAM pair in the anti-tumour immune response by T and NK cells.

Please clarify the paragraph about Galectin1/3 interaction and CD6 (lines 90 to 100): What I get is: Gal1/3 are ligands of CD6, but how does CD6 inhibit the galectin1/3 induced cell death (because it has a higher affinity and lead to a not-binding to the death inducing receptor? If yes, please add a sentence for explanation. thank you

Reply: As per reviewer’s indication a new explanative paragraph has been included in the new revised version of the manuscript as follows: “The mechanism by which CD6 surface expression negatively modulates galectin 1- and 3-induced T-cell death has not been explored in depth yet. Studies in a CD6-negative Jurkat T cell-derivative (2G5 cells) transfected with wild-type and cytoplasmic tail truncated CD6 isoforms indicate that such down-modulatory effect depends on integrity of the CD6’s cytoplasmic domain (Escoda-Ferran et al., 2014). This could relate to the previously reported fact that CD6 ligation protects from IgM-induced apoptosis in leukemic B-CLL cells by inducing an increased bcl-2/baxα ratio (Osorio et al., 1997)”

In this paragraph, I miss also a description of the signalling properties of CD6. How does this contribute to the multifacetted role? 

Reply: As per reviewer’s indication the following paragraph developing how the signaling properties of CD6 contribute to its multifaceted role has been included in the new revised version of the manuscript: “How the signaling capabilities of CD6 contribute to different immune response outputs is not fully understood. As stated above recent proteomic data shows that the CD6 signalosome behaves as a hub that assembles many different enzymes and adaptors, which can impact positively or negatively on signal propagation in different stimulation contexts and potentially drive to divergent outcomes (Mori et al., 2021).”

The last part (5) focusses more on CD166; maybe one should amend the title saying (CD6 and its binding partner CD166) and add also some (maybe) more general information on CD166. Or just "CD6 and its binding partners".

Reply: We do agree with the reviewer’s suggestion. Accordingly, the title of the new revised version of the manuscript has been modified as follows: “CD6 and its binding partners: newcomers to the block of cancer immunotherapies”.

For me it was not really clear which clinical studies are performed with anti-CD6 molecules. Maybe you could also add the registration numbers for the studies and/or a table listing them.

Reply: Following the reviewer’s indication, care has been taken to clearly state whether the studies reported have been performed with mAbs or other molecules throughout the new revised version of the manuscript.

In general: I am missing a lot of the murine studies, in which the function and potency of CD6 has been studied in detail.

Reply: Following the reviewer’s indication mouse studies in which the function of CD6 has been analyzed are described in more detail throughout the new revised version of the manuscript.

Please state more clearly in the respective paragraphs, if you are summarizing effect/functions/expression data of the human or the mouse system.

Reply: The mouse or human source of the experimental data reported has been clearly specified throughout the new revised version of the manuscript.

Please improve the resolution of the provided figure, it looks "blurry".

Reply: High resolution figures are now provided as separate files for final publication by editors.

Comments on the Quality of English Language: There are some minor misspellings and missing commata. Please correct :-)

Reply: The manuscript has been thoroughly and carefully revised for misspelling errors.

Reviewer 2 Report

Comments and Suggestions for Authors

The article titled ‘CD6: a Newcomer to the Block of Cancer Immunotherapies’ may be of interest in the context of a broad search for new targets for immunotherapy in oncological terms

Below are some comments on the article.

Not sure if the title is appropriate. Did the authors want to describe CD6 as a molecule that would block cancer immunotherapy? Or maybe they wanted to describe it as a potential therapeutic target for immunotherapy in cancer?

Itolizumab – should be written in lower caseNKG2A, NKG2D – należy podać peÅ‚ne nazwÄ™ w nawiasie

Line 175-176 …’ recombinant 175 human sCD6…’ Information on how it was recombined would be valuable.

Line 217-218 ‘The ideal targets for cancer therapies are tumour-specific antigens (TSAs), which are found on cancer cells only, but not on healthy cells.’ Adding information about what these molecules are and whether they are antibodies used in everyday clinical practice or clinical trials would be valuable.

line 221-224 ‘A similar situation applies to CEA (carcinoembryonic antigen), GPC3 (glypican-3), and PSA (prostate-specific antigen) for solid tumours of colorectal, hepatic or prostatic origin, respectively’ Information whether there are antibodies used in daily clinical practice or clinical trials against these molecules would be valuable.

A table summarizing antibodies/molecules targeting the CD6 pathway in anticancer treatment would be useful.

Author Response

Referee #2

Comments and Suggestions for Authors

The article titled ‘CD6: a Newcomer to the Block of Cancer Immunotherapies’ may be of interest in the context of a broad search for new targets for immunotherapy in oncological terms

Below are some comments on the article.

Not sure if the title is appropriate. Did the authors want to describe CD6 as a molecule that would block cancer immunotherapy? Or maybe they wanted to describe it as a potential therapeutic target for immunotherapy in cancer?

Reply: The authors intend to summarize the available information regarding the use of CD6 and its interacting partners as therapeutic targets for immunotherapy in cancer. Accordingly, and as per Reviewer #1 suggestion a new title is proposed: “CD6 and its binding partners: newcomers to the block of cancer immunotherapies”.

Itolizumab – should be written in lower case

Reply: Itolizumab has been changed to lower case.

Line 175-176 …’ recombinant 175 human sCD6…’ Information on how it was recombined would be valuable.

Reply: The term “recombinant” is used to indicate that human sCD6 is obtained by molecular biology methods and not by purification of the “naturally occurring” form found in human serum. The description of both forms can be found in references (Gimferrer et al., 2004) and (Sarrias et al., 2007) from the original version of the manuscript. This information is now provided in the new revised version of the manuscript.  

Line 217-218 ‘The ideal targets for cancer therapies are tumour-specific antigens (TSAs), which are found on cancer cells only, but not on healthy cells.’ Adding information about what these molecules are and whether they are antibodies used in everyday clinical practice or clinical trials would be valuable.

Reply: As per reviewer’s indication additional information on TSA has been included in the new revised version of the manuscript as follows: “TSA are neoantigens that can result from oncogenic viral infections and tumour-specific somatic mutations (normally single nucleotide variants but also frameshifts, splice variants, gene fusions or endogenous retroelements) (Smith et al., 2019). TSA can be common across different cancer patients and not present in the normal genome (shared TSA), thus having the potential to be used as broad-spectrum anticancer therapies. In other cases, TSA are unique and completely different from patient to patient (personalized TSA), and thus can only be specifically targeted to each patient (Zhang et al., 2021)”

line 221-224 ‘A similar situation applies to CEA (carcinoembryonic antigen), GPC3 (glypican-3), and PSA (prostate-specific antigen) for solid tumours of colorectal, hepatic or prostatic origin, respectively’. Information whether there are antibodies used in daily clinical practice or clinical trials against these molecules would be valuable.

Reply: We have rephrased the whole sentence to improve understanding by the readers as follows: “CD19, CD22, and BCMA (B-cell maturation antigen) are good examples of TAA for B-cell lineage malignancies, while CEA (carcinoembryonic antigen), GPC3 (glypican-3), and PSA (prostate-specific antigen) are for solid tumours of colorectal, hepatic or prostatic origin, respectively.” The requested additional information has not been included in the new revised version of the manuscript since it would exceed the purpose of our review.

A table summarizing antibodies/molecules targeting the CD6 pathway in anticancer treatment would be useful.

Reply: As per reviewer’s suggestion a new table summarizing the molecules targeting CD6 and its interacting partners in cancer immunotherapy has been included in the new revised version of the manuscript (see Table 1 below).

Table 1.  Summary of molecules (mAbs and engineered proteins) targeting CD6 and its interacting partners used in pre-clinical and clinical immunotherapy studies.

Molecule

Type

Study type

Disease

References

UMCD6

Humanised mouse anti-human CD6 mAb

Pre-clinical

CD166/ALCAM-expressing human tumours

(Ruth et al., 2021)

CD6-ADC

Humanised mouse anti-human CD6 mAb (UMCD6) toxin-conjugated

Pre-clinical

Human T-cell lymphomas

(Parameswaran et al., 2023)

CD6-CAR

CD6-based CAR-T cells

Pre-clinical

Human Colon adeno-carcinoma

(He et al., 2023)

CD166-CAR

Anti-CD166 CAR-T cells

Pre-clinical

Human Osteosarcoma

(Wang et al., 2019)

IOR-T1

Mouse anti-human CD6 mAb

Clinical

Human cutaneous T-cell lymphoma (CTCL)

(Hernández et al., 2016)

shCD6

Soluble human CD6 protein

Pre-clinical

Mouse tumours of different lineages

(Simões et al., 2020)

Anti-T12 (12.1)

Mouse anti-human CD6 mAb

Clinical

Human kidney transplant rejection

Human bone marrow transplant rejection

(Carpenter et al., 1983) (Reinherz et al., 1982)

Itolizumab

Humanised mouse anti-human CD6 (IOR-T1) mAb

Clinical (CTRI/2009/091/001009)

(RPCEC00000007 and

RPCEC00000035, Cuban Registry of Clinical Trials)

Psoriasis

Rheumatoid arthritis

(Anand et al., 2010; Krupashankar et al., 2014; Dogra et al., 2015) (Rodriguez et al., 2012; Rodríguez et al., 2018)